# Fertility after Curative Therapy for Sickle Cell Disease: A Comprehensive Review to Guide Care

**DOI:** 10.3390/jcm11092318

**Published:** 2022-04-21

**Authors:** Robert Sheppard Nickel, Jacqueline Y. Maher, Michael H. Hsieh, Meghan F. Davis, Matthew M. Hsieh, Lydia H. Pecker

**Affiliations:** 1Children’s National Hospital, Division of Hematology, Washington, DC 20001, USA; rnickel@childrensnational.org; 2Children’s National Hospital, Division of Blood and Marrow Transplantation, Washington, DC 20001, USA; 3School of Medicine and Health Sciences, The George Washington University, Washington, DC 20001, USA; mhsieh@childrensnational.org; 4Eunice Kennedy Shriver National Institute of Child Health and Human Development, Pediatric and Adolescent Gynecology, National Institutes of Health, Bethesda, MD 20810, USA; jacqueline.maher@nih.gov; 5Children’s National Hospital, Pediatric and Adolescent Gynecology Program, Washington, DC 20001, USA; 6Children’s National Hospital, Division of Urology, Washington, DC 20001, USA; 7Department of Urology, MedStar Georgetown University Hospital, Washington, DC 20001, USA; meghan.f.davis@medstar.net; 8Cellular and Molecular Therapeutics Branch, National Heart, Lung, and Blood Institute, National Institutes of Health, Bethesda, MD 20810, USA; matthewhs@nhlbi.nih.gov; 9Division of Hematology, Department of Medicine, School of Medicine, Johns Hopkins University, Baltimore, MD 20810, USA

**Keywords:** fertility, infertility, sickle cell disease, bone marrow transplant

## Abstract

Curative therapy for sickle cell disease (SCD) currently requires gonadotoxic conditioning that can impair future fertility. Fertility outcomes after curative therapy are likely affected by pre-transplant ovarian reserve or semen analysis parameters that may already be abnormal from SCD-related damage or hydroxyurea treatment. Outcomes are also likely affected by the conditioning regimen. Conditioning with myeloablative busulfan and cyclophosphamide causes serious gonadotoxicity particularly among post-pubertal females. Reduced-intensity and non-myeloablative conditioning may be acutely less gonadotoxic, but more short and long-term fertility outcome data after these approaches is needed. Fertility preservation including oocyte/embryo, ovarian tissue, sperm, and experimental testicular tissue cryopreservation should be offered to patients with SCD pursing curative therapy. Regardless of HSCT outcome, longitudinal post-HSCT fertility care is required.

## 1. Introduction

Fertility is a long-standing concern for individuals with sickle cell disease (SCD) and their families. The risk of infertility with SCD curative therapy is a barrier to patient acceptance and provider referral for hematopoietic stem cell transplantation (HSCT) [1,2,3]. Standard fertility preserving interventions for pre- and postpubescent females and postpubescent males exist, but best practices for integrating these interventions into SCD care are not established [4,5]. Exposure to gonadotoxic preparative regimens used to cure SCD is the clearest indication for fertility preservation. Infertility risk after curative therapy may vary by patient and regimen characteristics. Existing risk-stratification tools developed to counsel patients with cancer are not particularly helpful for risk-stratification because all HSCT patients are considered high risk for infertility regardless of the conditioning regimen [6]. Individuals and families considering curative SCD approaches may compare fertility outcomes to known and theorized risks to fertility associated with SCD and disease-modifying therapies for SCD. The purpose of this review is to provide a comprehensive overview of SCD-specific fertility risks with curative therapy. We describe fertility assessments, fertility considerations in pre- and post-transplant patients with SCD, and fertility preserving interventions for persons with ovaries and testes.

*Authors*’ *note on language*: There are two linguistic assumptions in this manuscript. We note that biologic sex at birth is not synonymous with gender identity. In this review we use “girls and women” interchangeably with people with ovaries and “boys and men” interchangeably with people with testes. A second language concern is in the use of the word fertility. Fertility, the ability to conceive children, is not easily studied. In a clinical sense, fertility is defined by the absence of infertility. Infertility is defined by the failure to conceive after 12 months of unprotected heterosexual intercourse. When infertility is a concern, for any reason, measures of ovarian reserve and semen analysis are used to assess fertility and predict success of fertility preserving interventions and guide the intervention. In this paper, we use “fertility measures” as synonyms for ovarian reserve testing and semen analysis. Hormone values are also important in the evaluation of potential ovarian insufficiency or testicular failure causing infertility. An overview of relevant measures of gonadal function and their significance in male and female physiology is provided in Table 1 and an overview of fertility preserving interventions in Table 2.

Fertility Considerations in Girls and Women.

**Table 1 jcm-11-02318-t001:** Measures of Gonadal Function.

Measure	Clinical Significance Women	Clinical Significance Men	Limitations
Follicle Stimulating Hormone (FSH)	Pituitary hormone that binds to ovarian granulosa cells where androgens are converted to estrogens Stimulates folliculogenesis Ovarian reserve marker ≥25–40 IU ×2, POI diagnosis	Pituitary hormone that stimulates spermatogenesis within Sertoli cells	Fluctuates with menstrual cycle May return to normal with time after chemoradiation Normal FSH does not guarantee spermatogenesis
Luteinizing Hormone (LH)	Pituitary hormone that stimulates progesterone and androgen production within ovarian theca cells Oocyte maturation—progresses from arrested prophase I to metaphase II (state required for fertilization)	Pituitary hormone that stimulates testosterone production within Leydig cells	Fluctuates with menstrual cycle
Estradiol	Produced from testosterone via aromatase in granulosa cells Breast and uterine development during puberty Uterine endometrial lining growth to prepare for embryo implantation Maintain bone mineral density Measure of ovarian function	Produced from testosterone via aromatase Testosterone deficiency can lead to elevated estradiol levels	Fluctuates with menstrual cycle May return to normal with time after chemoradiation
Progesterone	Stabilizes and maintains uterine lining for pregnancy Decline induces menses	n/a	Fluctuates with menstrual cycle
Testosterone	Hormone precursor for estradiol	Hormone critical in the male HPG axis	Deficiency associated with impaired spermatogenesis but not specific for impaired fertility
Antimullerian Hormone (AMH)	Ovarian reserve marker Helps predict ovarian response to IVF medications <1.1 ng/mL may indicate diminished ovarian reserve	n/a	Variable depending on age Large range of normal Only reflects pool of growing follicles May not reflect number of dormant primordial follicles May increase with time after chemoradiation
Antral Follicle Count (AFC)	Ovarian reserve marker Helps predict ovarian response to IVF medications and pregnancy rate	n/a	Inter and intra cycle variation Prone to observer bias
Inhibin B	Secreted by granulosa cells Negative feedback on FSH Possible marker of ovarian reserve	Protein secreted by Sertoli cells that inhibits FSH release from the pituitary	Abnormalities may signal dysfunction in the HPA axis Role in women remains unclear and controversial
Total motile sperm count	n/a	Calculation obtained by multiplying the volume of the ejaculate by the sperm concentration and the proportion of motile sperms divided by 100%	No cutoff is diagnostic for infertility

**Table 2 jcm-11-02318-t002:** Fertility preservation procedures.

Procedure	Patient Age	Clinical Use	Sickle Cell Specific Considerations
Ovarian tissue cryopreservation	Any age	Standard of care since 2019	Requires anesthesia, laparoscopy Risk for sickle cell crisis with surgery No consensus on whether to hold hydroxyurea or for how long
Oocyte or embryo cryopreservation	Post-menarcheal	Standard of care since 2013	Requires anesthesia / IV sedation Risk for Ovarian Hyperstimulation Syndrome, which may increase risk for sickle cell crisis No consensus on whether to hold hydroxyurea or for how long
Testicular tissue cryopreservation	Any age	Experimental	Requires anesthesia Fewer spermatogonial stem cells may impact future use
Sperm cryopreservation	Post-pubertal	Standard of care	Hydroxyurea treatment decreases the sperm count Unknown amount of time needed to hold hydroxyurea to improve sperm count Surgical sperm extraction may require sedation

## 2. Overview of Ovarian Reserve

Ovarian reserve is often assessed by measuring three values: Follicle Stimulating Hormone (FSH), Anti-Mullerian hormone (AMH), and antral follicle count (AFC). No single test is perfect, but the combination provides a good idea of ovarian function and egg supply.

Follicle Stimulating Hormone (FSH): Ovarian reserve can be assessed by measuring FSH along with estradiol levels on menstrual cycle day 2–4. Normal basal levels are FSH < 10 IU/L and estradiol <60 pg/mL [7].

Anti-Mullerian hormone (AMH): AMH is produced by granulosa cells of growing, immature ovarian follicles and is another measure of ovarian reserve. Age and pubertal stage guide interpretation of AMH values. In the general population, AMH decreases shortly after birth, increases in early childhood, plateaus, and then rises again until the mid-twenties when it peaks, and then declines non-linearly until perimenopause (40–50 years old) [8,9,10,11]. Unlike FSH, which fluctuates with the menstrual cycle, AMH may be measured at any time [12]. AMH is best validated as a predictor of response to controlled ovarian hyperstimulation in patients with infertility [13,14,15]. In people without a diagnosis of infertility, AMH is a poor predictor of spontaneous pregnancy [16]. However, low AMH occurs on a spectrum and very low levels are a risk for miscarriage and infertility [17]. AMH levels may be reported in several different base-unit scales and normal values are assay-dependent [18]. Low AMH levels are not a rational for withholding fertility preservation options from patients [19].

Antral follicle count (AFC): AFC is measured by ovarian ultrasound and is the total number of ovarian follicles on both ovaries measuring 2–10 mm. In women aged 18 to 24 years, the 50th percentile AFC is 14–17, and in women aged 35 to 40, it declines to 8–11 [20]. AFC < 7 is predictive of poorer response to controlled ovarian hyperstimulation [21].

Abnormal ovarian reserve: Abnormalities of ovarian reserve measures may be high or low. Very high AMH or AFC is not necessarily good; women with polycystic ovarian syndrome can have high AMH levels or AFC [22,23] and are at risk for infertility through anovulatory cycles, not a lack of oocytes. Low ovarian reserve before age 40 is classified as diminished ovarian reserve (DOR) or premature ovarian insufficiency (POI). DOR definitions vary. The modified Bologna Criteria provide a definition for poor ovarian response to hyperstimulation and includes women with regular menses, low AMH levels (<0.5–1.1 ng/mL), low AFC (<5–7), and/or elevated, but not menopausal FSH levels (10–25 mIU/mL) [24,25,26,27,28]. In pediatric studies, AMH under 5th percentile for age has been used to define abnormally low ovarian reserve [29], though the precise significance for future reproductive capacity is unclear in some contexts. DOR is a risk factor for miscarriage, infertility, poorer outcomes with controlled ovarian hyperstimulation, and early menopause [30]. More severely compromised ovarian reserve occurs in POI. POI is defined by age <40 years, amenorrhea for >4 months, and FSH levels in the menopausal range (≥25–40 mIU/mL) measured twice at least 1 month apart [31,32]. POI is an established, severe risk for infertility, with only a 5–10% chance of spontaneous pregnancy [32,33]. In the general population, about one percent of women <40 years old are affected [34]. No data on POI exists in SCD.

## 3. Ovarian Reserve in SCD

Women with SCD may have a narrower reproductive window than unaffected women [30,35,36]. Constitutional delay in puberty onset occurs commonly in girls with SCD, regular menstrual cycles are expected [37]. Normal ovarian reserve is reported in adolescents and young adults with untreated SCD [29,30], and sparse reports of ovarian follicle density in girls with SCD are, to date, normal [38]. However, SCD pathophysiologies, inflammation and hypoxic-ischemic injury, likely damage the ovaries [39]. The ovarian blood supply runs through the center of the ovary and there is progressively decreased vascularity in the outer layer of the ovarian cortex, which contains most dormant ovarian follicles. The idea that progressive vasculopathy of SCD may accelerate follicular atresia is a conclusion suggested by studies from the United Kingdom, United States, and Nigeria, showing that AMH declines faster in adult women with SCD than in unaffected women [30,35,40,41]. The lone study of menopause in SCD reported an earlier age at menopause than the general population, which also suggests that SCD accelerates age-associated decline in ovarian reserve [42].

As might be expected in a population with an accelerated decline in ovarian reserve, DOR occurs in some women with SCD. In three studies restricted to individuals with sickle cell anemia, DOR occurred in 27 of 102 women under age 31 years [29,30,40]. AFC in women with sickle cell anemia who were not pursuing fertility preservation is reported in one study of 26 women aged 18–31 years. Subjects with DOR defined by AMH criteria (*n* = 5) had a median AFC of 7 (IQR 7,7) compared to those without DOR median AFC was 12 (IQR 9, 19) [30]. The presence of DOR in some young women with SCD is an indication for broader fertility counseling and care, irrespective of pursuit of curative therapy [36].

Concerns about the effect of SCD treatments on ovarian reserve are focused on hydroxyurea. Small studies raise the possibility that hydroxyurea, a mild chemotherapeutic with unequivocal treatment benefits for children and adults with sickle cell anemia, impairs ovarian reserve. An analysis of DOR in young women with sickle cell anemia from the previously noted three independent cohorts identified that only hydroxyurea exposed subjects (27/84) had DOR [29,30,40]. Few subjects in these studies had no hydroxyurea exposure, precluding conclusions about unexposed patients. In the one cohort that conducted the analysis, SCD complications were not different between young women with and without DOR [30]. More evidence is needed to determine whether hydroxyurea is causally associated with DOR or a proxy for disease severity.

The effects of other chronic SCD treatments on ovarian reserve are not well studied. Iron overload is associated with chronic transfusion therapy and tends not to involve the pituitary in SCD, and no evidence yet establishes secondary hemochromatosis in SCD as a fertility risk [43,44]. Also, no evidence defines the effects of l-glutamine, voxelotor, or crizanlizumab. Individuals with SCD use many therapies for supportive care including non-steroidal anti-inflammatory drugs, opioids, and even marijuana. Polypharmacy for many patients contributes to the challenges of isolating the fertility effects of SCD and its disease modifying therapies.

This context is necessary for considering the effects of curative interventions for SCD several reasons. First, families may compare life-long hydroxyurea to HSCT; both may be indications for fertility preservation. Second, low pre-treatment ovarian reserve predicts poorer post-treatment ovarian reserve, so fertility in individuals with SCD may be more severely affected by gonadotoxic preparative regimens [45,46,47]. Third, even with AMH-defined DOR, AFC in young women with SCD may hover at the threshold for reasonable ovarian hyperstimulation outcomes, so fertility preserving interventions even for women with DOR may be successful if not deferred. Thus, guarded optimism regarding fertility preserving interventions is warranted.

## 4. Fertility Risks with Gonadotoxic Preparative Regimens for Girls/Women

In this section, we review available SCD-specific data from myeloablative, reduced intensity, and non-myeloablative regimens, noting limited outcomes are reported (Table 3). As in other end-organ assessments, baseline patient characteristics are needed to interpret post-HSCT risk: low baseline ovarian reserve may lead to more severe post-HSCT effects on ovarian reserve. Regimen specific risks also require consideration, but strong conclusions are challenging because limited data describes pre- or post-treatment fertility measures or other reproductive endpoints. Individuals with SCD are exposed to diverse HSCT regimens with variable alkylator doses [48]. Total body radiation (TBI) in the 2–4 Gy range is also used in some preparative regimens. While this is considered “low dose” TBI, ovarian radiation exposure of only 2 Gy is estimated to cause a 50% reduction in ovarian reserve [49]. The ovaries are particularly vulnerable to radiation as, unlike the testes, they are not readily shielded. TBI can also damage the uterus causing impaired uterine growth and distension, vascular damage, and impaired endometrial function [50,51]. This injury may increase the risk of spontaneous miscarriage, placental abnormalities, preterm delivery, and low birthweight infants [52,53,54]. Pubertal stage is also a consideration as the prepubertal uterus may be more sensitive to radiation. Limited data suggests that the risk for TBI associated uterine damage increases with radiation doses >/= 12 Gy. No SCD-specific evidence identifies the extent to which the uterus is affected by the 2–4 Gy radiation exposures used in some SCD HSCT regimens.

Historically HSCT for SCD was restricted to the pediatric setting with myeloablative busulfan and 200 mg/kg cyclophosphamide (Bu/Cy); thus, most existing post-HSCT gonadal function is reported in patients who received this regimen in childhood. Pre-HSCT fertility assessments are not reported in these populations, but long-term follow-up of participants in the first multicenter center HLA-identical sibling HSCT trial identified that 8/14 (57%) female subjects had evidence of POI and only 4/14 (29%) had normal estrogen levels. [55]. Despite evidence of serious gonadotoxicity in most female participants, two two pregnancies occurred in women at 13 and 14 years post-HSCT in this cohort.

European studies also reveal serious ovarian dysfunction after HSCT with Bu/Cy regimens and suggest this gonadotoxicity is lower for prepubertal girls. A finding that is similar to that described in girls with cancer [56,57,58,59,60,61,62]. A Belgian SCD study found that 7/10 girls age 6–14 at time of HSCT required pubertal induction or hormone replacement therapy (HRT) post-HSCT [63]. The three girls who did not require HRT were all prepubertal at the time of HSCT (age 2–10) and had spontaneous puberty. One of these prepubertal girls eventually gave birth. In a long-term follow-up of a large French cohort, older age at HSCT was associated with POI [64]. All post-pubertal subjects at time of HSCT (*n* = 14) had amenorrhea requiring HRT in the year after HSCT. In contrast, 9/32 (28%) of prepubertal subjects had spontaneous puberty. Girls who underwent spontaneous puberty were younger at HSCT than those requiring hormones for puberty induction (5.9 vs. 10.1 years). Among women older than 25 years at last visit in this cohort, 4/20 became pregnant—all were prepubertal at time of HSCT.

Less intensive HCST conditioning regimens are increasingly used in SCD [48]. The extent to which they better preserve short- or long-term fertility is unclear. In a study of ovarian reserve in SCD that included females treated with myeloablative Bu/Cy regimens at decreased Cy doses (120–180 mg/kg), all girls/women had very low AMH post-HSCT [65]. Reduced intensity conditioning may be less gonadotoxic [66]. In a study that included 22 pediatric female subjects (the number who were post-pubertal at time of HSCT is not reported) transplanted with alemtuzumab, fludarabine and melphalan (140 mg/m^2^), four adolescents resumed regular menstrual cycles [67]. Not all subjects developed amenorrhea. However more data addressing ovarian reserve and pregnancy attempts is needed to more substantively evaluate the potential gonadotoxicities of this regimen. A study of another reduced intensity regimen consisting of fludarabine, Bu 3.2 mg/kg, TBI 2 Gy, and Cy (29 mg/kg pre-HSCT, 66 mg/kg post-HSCT) in 22 adult women with SCD found that 15 (68%) became amenorrheic post-HSCT [68]. One spontaneous pregnancy occurred in this group, all women had low AMH values and 45% developed elevated FSH levels consistent with ovarian insufficiency. Studies of gonadal function with regimens that substitute thiotepa or a higher TBI dose for Bu are needed. Non-myeloablative conditioning using alemtuzumab and 3 Gy TBI may be less gonadotoxic than regimens with alkylating agents. Among 50 women with SCD transplanted with this approach, seven women reported pregnancies post-HSCT [69]. Gonadal function is under study in a multicenter clinical trial of pediatric patients with SCD transplanted with this regimen (NCT03587272).

**Table 3 jcm-11-02318-t003:** Summary of published fertility outcomes among patients with SCD after HSCT highlights the challenges in drawing definitive conclusion about fertility from existing reports. Methodological challenges include differences of pubertal stage at HSCT and variable HSCT-follow-up times, lack of pre-HSCT data, differences in reported outcomes, and lack of report of pregnancy attempts and/or infertility diagnoses. Data collection and reporting may be improved with adherence to the NHLBI’s new Cure Sickle Cell Initiative guidance on pre- and post-hSCT data collection.

Reference	Conditioning Regimen	Reported Fertility Outcomes		Comment on Source
		Female	Male	
Walters, M.C. *Biol. Blood Marrow Transplant.* 2010 [55]	Bu 14–16 mg/kg, CY 200 mg/kg	Amenorrhea N = 9/12 Spontaneous pregnancy/live births N = 2	Low testosterone N = 8/11	Gonadal function data reported in a subset of 22 female and 33 male survivors. At HSCT, subjects were <16 years and median age at last follow-up was 21 yrs.
Brachet, C. *J. Pediatr. Hematol. Oncol.* 2007 [63]	Bu 14–16 mg/kg, CY 200 mg/kg	Amenorrhea N = 7/10 Spontaneous pregnancy/live birth N = 1	Spontaneous puberty N = 9/9 Normal/low-normal testosterone N = 9/9 Azoospermia N = 1/2	At HSCT, subjects were <15 years. Median age at last follow-up 19 yrs. (female, N= 10), 17 yrs. (male, N = 9).
Bernaudin, F. *Haematologica* 2020 [64]	Bu ≥16 mg/kg, CY 200 mg/kg	Pre-pubertal at HSCT: Amenorrhea N = 23/32 Post-pubertal at time of HSCT: Amenorrhea, N = 14/14 4 women had 6 spontaneous pregnancies resulting in 5 live births OTC autograft pregnancy N = 1/2	Spontaneous puberty, normal testosterone in all, N = ? Fathered without IVF N = 3	Pregnancies restricted to women >25 yrs. at last f/u, N = 20. Reported male outcome restricted to boys who were pre-pubertal at HSCT and of pubertal age at last f/u (N not reported). In entire cohort of 125 males, only 19 men were age >25 yrs. at last f/u.
Elchuri, S.V. *J. Pediatr. Hematol. Oncol.* 2020 [65]	Bu 12.8–14 mg/kg, CY 120–200 mg/kg	Undetectable AMH N = 18/21, low AMH N = 21/21 Spontaneous pregnancy/live birth N = 1	Normal testosterone N = 16/16 No fathered pregnancies	At HSCT mean age 7.7 yrs. (female), 11 yrs. (male). At last f/u mean age 13.5 y (female), 22.1 yrs. (male).
Lukusa, A.K. *Pediatr. Hematol. Oncol.* 2009 [70]	Bu ^i^ CY ^i^ ± TLI	NA	Spontaneous puberty N = 5/5 Azoospermia N = 3/6 No fathered pregnancies	At HSCT, median age 12.5 yrs. Follow-up testing at median 28 yrs.
King, A.A. *Am. J. Hematol.* 2015 [67]	Alemtuzumab 48 mg, Flu 140 mg/m^2^, Mel 140 mg/m^2^	Regular menstrual cycles resumed in 4 adolescents	NA	Cohort of 22 females, without report of number post-pubertal or with amenorrhea
Zhao, J. *Pediatr. Transplant.* 2019 [71]	Alemtuzumab 48 mg, Flu 140 mg/m^2^, Mel 140 mg/m^2^	NA	Normal testosterone N = 3/3 Azoospermia N = 2/3 No fathered pregnancies	At HSCT, median age 14 yrs. F/u testing at median age 20 yrs.
Alzahrani, M. *Br. J. Haematol.* 2021 [69]	Alemtuzumab 1 mg/kg, TBI 3 Gy	7 women had spontaneous pregnancies	7 men fathered pregnancies without IVF	Entire cohort consisted of 50 females and 72 males. At HSCT median age 29 yrs, including 21 patients age ≥40 yrs. Median f/u 4 yrs. Total of 21 pregnancies with 18 live births and 3 elective abortions.
Boga, C. *Exp. Clin. Transplant.* 2022 [68]	Flu 150 mg/m^2^, Bu 3.2 mg/kg, TBI 2 Gy, CY 95 mg/kg	Amenorrhea N = 15/22 (pre-HSCT 0/22) Spontaneous pregnancy with miscarriage N = 1 IVF pregnancy using cryopreserved embryo with live birth N = 1	Azoospermia N = 14/19 (pre-HSCT 3/17) Fathered without IVF N = 1 Fathered with IVF N = 1	At HSCT all patients age >18 yrs, mean age 29 yrs. At last f/u mean age 33 yrs. Only 10 women and 6 men were married post-HSCT.

Bu—busulfan. CY—cyclophosphamide. TLI—total lymph node irradiation. TBI—total body irradiation. Flu—fludarabine. Mel—melphalan. AMH—anti-Mullerian hormone. yrs.—years. f/u—follow-up. IVF—in vitro fertilization. NA—not applicable. ^i^ dose not reported.

## 5. Female Fertility Preservation

Oocyte or embryo cryopreservation: Controlled ovarian hyperstimulation requires 10–14 days and includes close follicle monitoring with transvaginal ultrasounds and blood tests, multiple days of injected ovarian simulation medications, and anesthesia for a minimally invasive transvaginal oocyte retrieval [72]. Oocytes can be cryopreserved or fertilized and grown to day 5 embryos for cryopreservation. No SCD-specific data informs the ideal number of oocytes to preserve and prediction is complicated in women with SCD. In women without SCD less than 35 years old, the chance of live-birth from 5 oocytes is 15.4% and from 10 oocytes is 60.5% [73,74], so, if possible, freezing 15–20 oocytes for those <38 years is recommended [75]. Additionally, women with SCD whose partner has a beta hemoglobinopathy trait may be interested in using in vitro fertilization with pre-implantation genetic testing for monogenic diseases (PGT-M) to prevent SCD in offspring [76]. This typically requires more oocytes to have a euploid embryo without SCD to transfer and might reduce the available number of embryos for transfer. Furthermore, if TBI is included in the HSCT regimen, the uterus may be affected by radiation, further affecting chances of successful implantation. Until disease-specific data is available, conservative approaches that safely maximize oocyte retrieval are indicated.

Ovarian tissue cryopreservation (OTC): In 2019 OTC became a standard fertility preserving option for patients receiving gonadotoxic treatment [5]. OTC is sometimes performed in women without time to go through a controlled ovarian hyperstimulation cycle and retrieval to cryopreserve oocytes. Since HSCT in SCD is never an emergency, we expect this to be an uncommon use of OTC. More commonly, it will be used for prepubertal girls for whom OTC is also the only fertility preservation option since prepubertal ovaries do not meaningfully respond to controlled ovarian hyperstimulation stimulation mediations. OTC requires anesthesia and surgery, typically performed laparoscopically. Standard pre-operative interventions for individuals with SCD are used. OTC pregnancies are reported with over 360 transplantations and over 130 live births worldwide [74,77,78,79,80]. OTC autografting can also be used to successfully induce puberty in girls after HSCT [81]. There is little SCD-specific OTC pregnancy data but there are at least two reports: (1) OTC performed at age 14 years (post-pubertal but pre-menarcheal) pre-HSCT with ovarian tissue transplantation at age 24, resulting in a spontaneous pregnancy and live birth after 2 years; (2) OTC performed at age 20 years pre-HSCT with ovarian tissue transplantation at age 23, resulting in two spontaneous pregnancies and live births after 6 months and 3 years later [64,82,83].

SCD-specific concerns: In both methods of fertility preservation, coordination between care teams is necessary due to unique procedural risks in SCD. Oocyte cryopreservation is a risk for ovarian hyperstimulation syndrome (OHSS). OHSS is a capillary leak syndrome that is graded by severity of associated complicates. OHSS is characterized by hypercoagulability, electrolyte abnormalities, ascites, anasarca, pulmonary edema and respiratory distress, liver dysfunction, and acute renal insufficiency. Clearly these complications are especially undesirable for women with SCD who have baseline renal impairment, hypercoagulability and may have a personal history of venous thromboembolic events. Oocyte cryopreservation complications described in women with SCD include interruption of hyperstimulation cycle due to pain crisis, and life-threatening acute chest syndrome [84,85,86]. Ovarian stimulation protocols are individually tailored to reduce OHSS risks; GnRH agonist trigger may be indicated [86]. Reducing peri-procedural and OHSS risks require multi-disciplinary collaboration between specialists in reproductive endocrinology and infertility (REI), anesthesiologists, and hematologists with SCD expertise. A published clinical algorithm includes considerations for the use of pre-stimulation red cell transfusion and considerations for prophylactic anticoagulation to inform this multidisciplinary care [86].

Ovarian hyperstimulation medication regimens are complex and high stakes. As many individuals with SCD have overt or insidious cognitive impairment, ensuring that a care partner is included in clinical teaching and at-home care can help address complexities of care including medication timing and administration [87,88].

Outcomes: Case series of oocyte cryopreservation in girls and women with SCD raise concern that at least some subjects have suboptimal oocyte yield for their age. Healthy egg donors with an average age of 26.3 years have an average of 24.6 oocytes retrieved per stimulation cycle [89]. These numbers are lower in small studies of young women with SCD. In one study, among eight SCD adolescent girls 14–18 years who underwent oocyte cryopreservation before HSCT, a mean of 12.1 oocytes were preserved; four girls had ≤7 mature oocytes [90]. Among five patients 15–32 years old, four had a mean of 12.5 oocytes cryopreserved and one cryopreserved seven embryos [86]. Finally, among six patients 20–38 years old, a mean of 10.7 oocytes were cryopreserved, outcomes similar to patients with immune deficiencies (GATA2 or DOCK8 deficiency) [91]. These series do not analyze complete ovarian reserve measures nor SCD-treatment exposure. Hydroxyurea exposed subjects have successfully undergone oocyte / embryo cryopreservation [86]; however, whether treatment affects oocyte quality, oocyte fertilization, embryo maturation or pregnancy outcomes is not established, and evidence addressing this concern is limited to murine studies [92,93]. Systematic, multidisciplinary studies are needed to optimize fertility preservation and assess pregnancy outcomes.

Controversial evidence for alternative fertility preserving interventions: Gonadotropin releasing hormone agonists (GnRHa), including leuprolide, triptorelin, and goserelin, may help with irregular and heavy bleeding during HSCT and possibly decrease the risk for POI. GnRHa for ovarian protection is controversial as study results are mixed [94,95,96,97,98,99,100,101]. Consequently, GnRHa is not a substitute for pre-transplant fertility preserving interventions [102,103,104], but may be offered along with cryopreservation techniques or if fertility preservation is not feasible.

## 6. Monitoring and Recommendations Post-Curative Therapy for Girls/Women

Long-term follow-up in clinics with expertise in screening and treating late effect of exposure to chemotherapy and radiation are essential for individuals with SCD who pursue curative interventions, regardless of whether they are cured of SCD. This follow-up should include sexual and reproductive care because complications may arise and because fertility preservation may also be offered in the post-transplant setting. For individuals transplanted in early childhood without fertility preservation, regular ovarian reserve testing and providing counseling about ongoing fertility risks are indicated. Patients with decreasing AMH and rising FSH levels or irregular menses should be referred for specialize fertility care.

The National Heart, Lung and Blood Institute’s Cure Sickle Cell Initiative provides standards for hormone and fertility assessments before and after gonadotoxic regimen exposure [105]. Testing includes AMH, AFC, FSH, LH, and estradiol before exposure and annually thereafter since ovarian function may fluctuate after HSCT [63]. The trajectory of post-HSCT gonadal hormone recovery varies and exogenous hormone administration may be required. 

Prepubertal girls who do not initiate spontaneous pubarche require pediatric endocrinology specialty care with pubertal induction around 11–12 years. Estradiol is started and progesterone added approximately two years later or with the first episode of vaginal bleeding for uterine protection. In peripubertal girls with stalled puberty, HRT may also be indicated. 

Postmenarcheal girls should be screened for vaginal dryness and discomfort, which may reflect graft-versus-host disease or hypoestrogenic vaginal atrophy. For bone, cardiovascular, cognitive health, and sexual health, women with POI usually continue HRT through age 50 years, the average age of natural menopause in women without SCD [106,107]. Since women with POI after HSCT still have a chance of spontaneous pregnancy [65], a contraceptive form of HRT such as a continuous combined oral contraceptive pill is indicated if pregnancy is not desired. If TBI is included in the HSCT regimen, an ultrasound with Doppler to assess uterine volume, vasculature, and endometrial thickness can be performed [108].

Finally, all individuals with SCD post-HSCT, whether they underwent fertility preservation or not, require formal genetic counseling regarding inheritance of hemoglobinopathy traits and affirming that even individuals cured of SCD will pass a hemoglobinopathy trait to their offspring. Genetic counseling can clarify this risk, inform partner testing, and facilitate reproductive decision making including the option of preimplantation genetic testing of embryos conceived via in vitro fertilization (IVF) [76,109].


Fertility considerations in boys and men.


## 7. Overview of Semen Analysis

For men, fertility evaluation includes FSH, LH, and testosterone measures and semen analysis (SA). The World Health Organization SA parameters were developed from semen analyses of men whose partners became pregnant within 12 months or less with the cut offs representing the 5th percentile (Table 4) [110]. Another measure used is the total motile sperm count (TMSC) which has a higher correlation with fertility than WHO semen parameters, but is not yet included in American Society for Reproductive Medicine guidelines as part of the initial evaluation [111]. With few exceptions, no single SA abnormality is diagnostic for infertility [112]. At the same time, a normal SA does not guarantee fertility as up to 25% of men with infertility have a normal SA [113]. However the risk of infertility increases with an increasing number of SA parameters in the subfertile range [114]. SA samples may vary, so more than one SA may be required to draw meaningful conclusions [112].

## 8. Fertility in Men with SCD

SCD damages the testes which, like the ovaries, are exposed to inflammation and hypoxic-ischemic damage. This injury impairs normal spermatogenesis and causes testicular infarction [115,116,117]. There are high rates of abnormal semen analyses in men with SCD taking no disease modifying therapy [115,118,119,120,121], but sperm counts range from normal to frank azoospermia [121]. Additionally, over 30% of men with SCD experience priapism which can cause erectile dysfunction [122,123,124]. Erectile dysfunction does not affect spermatogenesis, but is a fertility barrier. Further, some treatments for priapism may impair fertility, reduce libido, and exacerbate erectile dysfunction. Finally, men with SCD have mild hypogonadism compared to unaffected men [125,126].

Limited evidence defines the effects of SCD treatments on male fertility. This literature primarily concerns the adverse effects of untreated SCD and hydroxyurea. Hydroxyurea treatment decreases sperm counts as demonstrated in a prospective study of 35 men with SCD [127]. Six months after starting hydroxyurea the median total sperm count decreased from 61.6 to only 0.63 million, the proportion of subjects with an abnormal total sperm count increased from 40% to 86%, and six treated men became azoospermic. The extent to which this effect is reversible is unclear. In various studies, hydroxyurea discontinuation led to sperm count improvements in some, but not all men [121,128,129,130]. A cross-sectional study of young men with SCD who were never treated with hydroxyurea (*n* = 23) and previously treated with hydroxyurea during childhood (*n* = 15) found no significant differences in SA parameters between the two groups [131]. Two studies evaluated testicular tissue from boys with SCD who underwent biopsy for fertility preservation prior to HSCT. The first compared boys never exposed to hydroxyurea (*n* = 13), previously treated with hydroxyurea (*n* = 11), and currently receiving hydroxyurea (*n* = 6) and found no significant differences in testicular tissue [132]. In contrast, the second found that an earlier age of hydroxyurea initiation was associated with a decreased number of spermatogonia [133]. In the pre-HSCT setting, these results suggest that fertility preserving interventions may be contingent on baseline SCD injury and the extent to which hydroxyurea effects are reversed upon treatment discontinuation.

Other medications commonly used by people with SCD may also affect male fertility. Opioids and marijuana may disrupt the hypothalamic–pituitary–adrenal (HPA) axis and negatively affect male fertility [134]. Limited evidence suggests chronic NSAID use does not affect the HPA axis [135]. Studies of disease-modifying medications other than hydroxyurea on SCD male fertility are lacking, although a study of crizanlizumab to prevent priapism is ongoing (NCT03938454).

## 9. Fertility Risks with Gonadotoxic Preparative Regimens for Boys/Men

Limited post-cure fertility and pregnancy related data exist for men with SCD (Table 3). Most reports do not provide pre-transplant fertility assessments in males with SCD. This is problematic since both SCD and hydroxyurea can affect semen parameters.

The existing data mostly concerns outcomes after HSCT with myeloablative Bu/Cy regimens. Long-term follow-up of participants in the first multicenter center HSCT trial for SCD identified that among male participants, none had elevated LH/FSH levels, but 4/13 evaluated patients (31%) had low levels LH/FSH (clinical significance unclear) and 10/13 (77%) had low testosterone levels [55]. In contrast, a French cohort involving 125 male patients reported that all had normal testosterone, FSH, and LH levels [64]. Of note, unlike girls who commonly require HRT after transplant with Bu/Cy, prepubertal boys can be expected to have spontaneous puberty post-HSCT [63,64]. Nonetheless, impaired spermatogenesis may occur despite normal pubertal development. In a Belgian cohort, SA was abnormal (azoospermia and oligio-teratospermia) in the two young men who post-HSCT had this testing [63]. A comparison of SA parameters in 6 men post-HSCT to 4 men on hydroxyurea identified equal azoospermia rates in both groups (50%) [70]. Of note, post-HSCT patients with azoospermia in this small study received total lymph node irradiation as part of their HSCT in addition to Bu/Cy. Two men in the post-HSCT group had a total sperm count >50 million compared to a maximum sperm count of only 2 million in the hydroxyurea group. At least some men have preserved fertility after myeloablative Bu/Cy is possible. The previously noted French cohort reported that three of 19 men over age 25 years old fathered children spontaneously.

Some results of reduced intensity HSCT conditioning regimens are available. In a study comparing reduced dose Bu/Cy regimens to conventional doses [65], measured male endpoints identified normal post-HSCT testosterone and LH. Reduced intensity conditioning may be theoretically less gonadotoxic; however, spermatogenesis was impaired in a report of three young men with SCD treated with alemtuzumab, fludarabine, and melphalan (140 mg/m^2^) [71]. Two men had post-HSCT azoospermia and one had a total sperm count of only 0.35 million. The absence of pre-HSCT semen analysis for these individuals precludes definitive conclusions about the regimen’s gonadotoxicity. In a study of 21 men with SCD transplanted with fludarabine, Bu 3.2 mg/kg, TBI 2 Gy, and Cy (29 mg/kg pre-HSCT, 66 mg/kg post-HSCT), azoospermia occurred in 17% of men pre-HSCT and 74% post-HSCT, demonstrating that reduced intensity conditioning can still seriously impair spermatogenesis [68].

Nonmyeloablative conditioning using alemtuzumab and 3 Gy TBI may be less gonadotoxic especially for boys/men given irradiation testicular shielding. Among the 72 men with SCD transplanted with this approach, 7 men reportedly fathered children [69], including one man who had pre-HSCT gonadal dysfunction and received hormone treatment post-HSCT to conceive [136]. An ongoing analysis will include hormone levels, semen analysis parameters, and reproductive health questionnaire results from some of these subjects.

## 10. Fertility Preservation

Sperm cryopreservation: Sperm cryopreservation is used to preserve fertility in post-pubertal males. No studies specifically report on sperm banking procedures and subsequent reproductive outcomes among men with SCD. TMSC greater than 9 million optimizes pregnancy rates for intrauterine insemination (IUI), but pregnancy with IUI can be achieved with much lower TMSCs (<250,000) [137]. For those with lower sperm counts, sperm can be cryopreserved and used for IVF with intracytoplasmic sperm injection (ICSI) [138]. Since men with SCD often have abnormal sperm counts, they may require multiple collections to bank an adequate sample to optimize future fertility outcomes [139]. Sperm extraction procedures, such as testicular sperm extraction (TESE), are options for post-pubertal patients unwilling or unable to ejaculate and for some patients with azoospermia. Micro-TESE is a care standard for men with non-obstructive azoospermia [138].

Hydroxyurea’s negative effect on sperm count may impede sperm banking [140,141]. When possible, if desired by the patient, we stop hydroxyurea and initiate chronic transfusion while awaiting sperm recovery. Suspending hydroxyurea treatment is challenging, especially in people with severe disease manifestations and when chronic transfusion is not possible. The minimum time necessary for sperm recovery is unknown [131], but in our experience months, rather than weeks, are required. Most men going through this process have not had pre-hydroxyurea SA performed, so predicting the extent to which recovery will occur after stopping hydroxyurea is difficult. Since HSCT conditioning regimens may include pre-HSCT hydroxyurea treatment to potentially enhance donor engraftment [142]. Early discussions and access to fertility preservation is warranted for individuals pursuing cure.

Testicular tissue cryopreservation: For prepubertal boys and patients unable to sperm bank, testicular tissue cryopreservation (TTC) is an experimental fertility preserving intervention currently offered at a limited number of centers. TTC requires unilateral testicular biopsy to obtain spermatogonial stem cells that are cryopreserved and may be used to produce mature sperm. Sperm maturation methods are under study with no successful pregnancies yet reported. Hydroxyurea may not damage spermatogonial stem cells, so may be continued, but in the absence of definitive data, this decision should involve shared decision making [132]. Males with SCD have, on average, fewer spermatogonial stem cells than healthy boys [132,133,143]; but whether this will meaningfully impact outcomes is unknown. A recent review includes a useful overview of TTC in SCD [144].

The TTC procedure involves testicular biopsy and sedation or general anesthesia. Typically, boys undergo the latter due to inability to tolerate sedation. The procedure takes under one hour. In the general population, testicular biopsy is low risk (2–3%) with minor procedural complications reported [145,146,147,148,149,150,151,152,153,154]. SCD-associated anesthesia risks and differential post-procedure pain response requires consideration [155], but no SCD data specific to this procedure exists. Testicular biopsy can be paired with another procedure, such as central line placement, in the immediate pre-HSCT time period, but a procedural complication such as hematoma or skin infection may delay initiation of HSCT. More conservatively, testicular biopsy may be scheduled a few weeks prior to HSCT even if it requires a separate hospitalization and sedation. In those without SCD, one year follow-up after unilateral testicular biopsy for cryopreservation found no significant differences in the size of biopsied versus non-biopsied testis [150]. More research is needed, but this suggests that, depending on the volume of testicular tissue harvested, the biopsy may not impair testicular growth and development. Future studies will define outcomes of post-HSCT reimplantation of testicular tissue, including pregnancy.

## 11. Monitoring and Recommendations Post-Curative Therapy for Boys/Men

For boys/men, we suggest annual post-HSCT testing to include FSH, LH, and testosterone. Since men with normal hormone levels can still have impaired spermatogenesis, SA testing should also be done to evaluate gonadal function. Men with azoospermia early post-HSCT may not be infertile as spermatogenesis recovery can occur many years after HSCT [156]. Men should thus be counseled that contraception is indicated if pregnancy is not desired even if they have documented azoospermia. Additionally, current artificial reproductive technology includes the possibility of using ICSI enabling some men with abnormal SA parameters and very low sperm counts as a possible cause of infertility to conceive with IVF. Finally, as discussed in the post-HSCT monitoring section for girls/women, all patients after curative therapy for SCD should receive genetic counseling since they will pass on a hemoglobinopathy trait.

## 12. Conclusions

Less toxic curative approaches may make fertility preservation unnecessary in the future, but at present, fertility preservation is a fundamental component of SCD care. There are many uncertainties associated with counseling about fertility preservation for SCD. These include, but are not limited to, precise fertility risks with preparative regimens, pregnancy outcomes with preserved gametes, and spontaneous pregnancy rates. Incorporating this uncertainty into counseling is necessary since definitive data will take time to accrue. Given the potential for gonadotoxicity, even with non-myeloablative conditioning, fertility preservation is indicated for all patients with SCD pursuing curative therapy.

The lack of universal access to fertility preservation before curative therapy is a medical, ethical, and practical concern. Many states have introduced bills to mandate insurance companies to cover fertility preservation procedures for patients who are facing potential infertility as a result of medical treatment (iatrogenic infertility). As of October 2021, eleven states mandate insurance coverage for fertility preservation. However, even states with fertility mandates limit coverage contingent on insurer and insurance type. Public and federal insurance do not cover fertility preservation except in Illinois where a public insurance program covers fertility preservation [157]. Marked disparities in access to fertility counseling and treatment for individuals with SCD are especially stark when compared to individuals with cancer [4,158]. At the National Institutes of Health, subjects participating in research protocols that require exposure to gonadotoxic regimens are offered fertility preservation. This precedent-setting approach needs consideration for all foundation- and government-sponsored research.

To provide indicated the pre- and post-HSCT fertility care to individuals with SCD, clinical care structures need immediate reform, hematologists and HSCT specialists need to engage fertility specialists as essential SCD care partners, and policy changes are needed to resolve unequal access to fertility care in SCD [36].

## Figures and Tables

**Table 4 jcm-11-02318-t004:** WHO Semen Analysis Parameters.

Parameter	One-Sided Lower Reference Limit (5th Percentile with 95% Confidence Interval)
Semen Volume	1.5 mL (1.4–1.7)
Sperm Concentration (million/mL)	15 × 10^6^ (12–16 × 10^6^)
Total Sperm Count (per ejaculate)	39 × 10^6^ (33–46 × 10^6^)
Vitality	58% (55–63%)
Total Motility (Progressive and non-progressive)	40% (38–42%)
Morphologically Normal Forms	4% (3–4%)

## Data Availability

Not applicable.

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
