# Peer review of "Fertility after Curative Therapy for Sickle Cell Disease: A Comprehensive Review to Guide Care"

_jcm, 2022, doi:10.3390/jcm11092318_

Round 1

Reviewer 1 Report

Dear Authors,

Thank you very much for this very comprehensive  review on the fertility of sickle cell patients after transplant.

My only suggestion would be to summarize all reported live birthcases in a table (specifying conditionning regimen, hormonal substitution, OTC, reference, ...).

Author Response

Dear Authors,

Thank you very much for this very comprehensive review on the fertility of sickle cell patients after transplant.  My only suggestion would be to summarize all reported live births in a table (specifying conditioning regimen, hormonal substitution, OTC, reference, ...).

Thank you for this suggestion. We added Table 2 which reviews the existing data and in so doing also highlights the many deficiencies in reported outcomes.

Reviewer 2 Report

Fertility is a major issue in transplant for SCD. This review, is a good overview summarizing the most questions and answers concerning this topic and will be an useful tool for physicians.  

Minor remarks:

Lines 185-186: “ In a study of ovarian reserve in SCD that included females treated with reduced dose Bu (12.8 mg/kg) / Cy (120-180 mg/kg) regimens, all girls/women had very low  AMH post-HSCT” . Only cyclophosphamide dose is reduced , Bu at 12.8mg/kg is IV equivalent of oral 16mg/kg.

At the end of the section 4: Fertility risks with gonadotoxic preparative regimens for girls/women . Authors should to add data concerning haplo transplant after RIC approach according the Hopkins protocol with intensification by adding thiotetepa or increasing dose of NMA TBI. They should briefly discuss how these conditioning can impact post-transplant fertility .  

Line 210: Further-209 more, if TBI is included in the HSCT regimen, the uterus may be affected by radiation, 210 further affecting chances of successful implantation. Do the authors have any data about impact of low dose TBI on uterus?

Line 226: this section could be completed by experience reported by Poirot, Acta Obste, Gyneco, Scandi 2019, Poirot, Lancet 2012, Donnez Human Reprod 2006.

Line 240: It would be useful to detail the management of these patients : anticoagulation, reduce HbS etc ..

Could authors give or discuss recommandations according to the modalities of transplant in female : in cases of MAC (BUbased) , NMA (alemtuzumab/3Gy TBI) , or RIC “ baltimore haplo approach with addition of thiotepa or 3/4GY TBI. What we know, and what we don’t know

Author Response

Fertility is a major issue in transplant for SCD. This review, is a good overview summarizing the most questions and answers concerning this topic and will be an useful tool for physicians.  

Thank you

Minor remarks:

Lines 185-186: “ In a study of ovarian reserve in SCD that included females treated with reduced dose Bu (12.8 mg/kg) / Cy (120-180 mg/kg) regimens, all girls/women had very low  AMH post-HSCT” . Only cyclophosphamide dose is reduced , Bu at 12.8mg/kg is IV equivalent of oral 16mg/kg.

This sentence is revised and now reads:

In a study of ovarian reserve in SCD that included females treated with myeloablative Bu/Cy regimens at decreased Cy doses (120-180 mg/kg) , all girls/women had very low AMH post-HSCT.60 

At the end of the section 4: Fertility risks with gonadotoxic preparative regimens for girls/women. Authors should to add data concerning haplo transplant after RIC approach according the Hopkins protocol with intensification by adding thiotepa or increasing dose of NMA TBI. They should briefly discuss how these conditioning can impact post-transplant fertility .  

Thank you for this suggestion. We wrote:

In a study that included 22 pediatric female subjects (the number who were postpubertal at time of HSCT is not reported) transplanted with alemtuzumab, fludarabine and melphalan (140mg/m2), 4 adolescents resumed regular menstrual cycles.67 That not all subjects developed amenorrhea is encouraging, however additional information about ovarian reserve and pregnancy attempts is needed to more substantively evaluate the potential gonadotoxicities of this regimen. A study of another reduced intensity regimen consisting of fludarabine, Bu 3.2 mg/kg, TBI 2 Gy, and Cy (29 mg/kg pre-HSCT, 66 mg/kg post-HSCT) in 22 adult women with SCD found that 15 (68%) became amenorrheic post-HSCT.68 While one spontaneous pregnancy occurred in this group, all women had low AMH values and 45% developed elevated FSH levels consistent with ovarian insufficiency. Further study of gonadal function with regimens that substitute thiotepa or a higher TBI dose for Bu are needed.

Line 210: Further-209 more, if TBI is included in the HSCT regimen, the uterus may be affected by radiation, 210 further affecting chances of successful implantation. Do the authors have any data about impact of low dose TBI on uterus?

Thank you for this suggestion. We amended the manuscript which now reads:

TBI can also damage the uterus causing impaired uterine growth and distension, vascular damage, and impaired endometrial function. This may increase the risk of spontaneous miscarriage, placental abnormalities, preterm delivery, and low birthweight infants. Pubertal stage is a consideration as the prepubertal uterus may be more sensitive to radiation. Limited data suggests that uterine damage is more likely with radiation doses >/= 12Gy. No SCD-specific evidence assessing the extent to which the 2 - 4 Gy radiation exposure required in some SCD HSCT regimens affects the uterus is available. 

Line 226: this section could be completed by experience reported by Poirot, Acta Obste, Gyneco, Scandi 2019, Poirot, Lancet 2012, Donnez Human Reprod 2006.

Thank you, the citations are added.  

Line 240: It would be useful to detail the management of these patients : anticoagulation, reduce HbS etc.

Thank you. We revised this section and added a sentence to address this management and to direct readers to a comprehensive paper on this topic. Unfortunately, no rigorous data supports a particular management approach.

Could authors give or discuss recommendations according to the modalities of transplant in female : in cases of MAC (BUbased) , NMA (alemtuzumab/3Gy TBI) , or RIC “ baltimore haplo approach with addition of thiotepa or 3/4GY TBI. What we know, and what we don’t know.

We understand this comment to be a request to tailor fertility preservation or longitidunal follow-up recommendations for fertility preservation based on transplant regimen. Unfortunately, we cannot provide this level of nuance to our recommendations. In the future, systematic data which includes pre-HSCT and long-term post-HSCT fertility measures may enable nuanced appraisal of regimen specific effects on female fertility and their relative indication for fertility care. At present, we know that women with SCD have low ovarian reserve at baseline, that alkylating agents are considered high risk for infertility and that even low-dose TBI can meaningfully reduce ovarian reserve. Thus, for females with SCD, these exposures all pose infertility risks and long-term care is required. Until rigorous data suggesting otherwise is available, we recommend a conservative standard that all individuals with SCD who have not completed their families are considered at risk for infertility in the setting of HSCT and should receive pre- and post-HSCT fertility care. We make this recommendation mindful of painful history of involuntary sterilization among members of the U.S. African American community, the need for trusting relationships between individuals with SCD and their care teams, and in accordance with the NHLBI’s Cure Sickle Cell Disease Initiative recommendations.

This manuscript is a resubmission of an earlier submission. The following is a list of the peer review reports and author responses from that submission.